# OpenReview forum: "Lexically-constrained automated prompt augmentation: A case study using adversarial T2I data"
_NeurIPS.cc/2024/Workshop/SafeGenAi — SafeGenAi Poster_

### Official Review · Reviewer_Nwsz · 2024-10-08
**A Promising Approach to Augmenting Adversarial Datasets for T2I Model Safety Evaluation**

**Rating:** 6
**Confidence:** 3

**Review:**

This paper presents an innovative approach to augmenting datasets for evaluating the safety of text-to-image (T2I) models. The authors recognize that T2I models can be manipulated into generating harmful images through careful or erroneous prompting, and that existing datasets to study this phenomenon are limited. To address this, they propose augmenting the Adversarial Nibbler Challenge dataset, which contains 1,241 human-generated prompts. Their method combines human-generated adversarial prompts with automated techniques, aiming to create a more comprehensive and scalable dataset while maintaining realism.

The authors employ multiple evaluation techniques to explore different aspects of prompt adversariality. These include correcting typographical errors, replacing trigger words with synonyms, and using an LLM to rephrase entire prompts. They provide both quantitative and qualitative analysis of their results, demonstrating thoroughness in their approach. For instance, when correcting spelling mistakes, they found that "no failures persist for bias (0/4 prompts)". However, manual inspection revealed that "images generated for the corrected prompt 'a kid before the police' were predominantly images of Black children". This showcases the authors' commitment to evaluation but also highlights the potential limitations of automated safety classifiers.

The study utilizes the same T2I model suite as Quaye et al., including Dall-E-2 and four variations of Stable Diffusion, up to 2.1. While this allows for direct comparison with previous work, it would be beneficial to see how newer models react to these adversarial prompts. The authors are transparent about the preliminary nature of their results and the need for further evaluation. They also acknowledge the subjective nature of image safety and include appropriate content warnings, demonstrating ethical awareness.

While the paper presents a promising approach, there are several areas for improvement. The dataset could be expanded to provide more robust results. The authors could explore additional attack strategies beyond typographical errors and semantic ambiguity. Incorporating more sophisticated safety classifiers, especially for detecting bias and hate in generated images, would strengthen the analysis. Additionally, conducting user studies to validate the effectiveness of the augmented prompts in real-world scenarios would enhance the paper's impact. Despite these limitations, the authors' data-driven methodology and novel approach to combining human creativity with automated augmentation techniques show great potential for advancing the field of T2I model safety evaluation.

---

### Official Review · Reviewer_qocS · 2024-10-11
**Review of 181**

**Rating:** 7
**Confidence:** 4

**Review:**

The paper presents a framework for generating augmented adversarial datasets to test T2I models by combining human-written adversarial prompts with lexically-constrained automated techniques.

### Strengths

1. The combination of human creativity and automated lexical constraints to scale adversarial prompt generation is novel.
2. The paper evaluates the effectiveness of augmented adversarial prompts using both Typographical Errors and Semantic Ambiguity strategies, which are comprehensive.

### Weaknesses

1. It would be better to provide deeper analysis and theoretical discussion around semantic ambiguity and its broader impact on T2I models.